# Introduction of Electron Donor Groups into the Azulene Structure: The Appearance of Intense Absorption and Emission in the Visible Region

**DOI:** 10.3390/molecules29143354

**Published:** 2024-07-17

**Authors:** Nurlan Merkhatuly, Ablaykhan Iskanderov, Saltanat Abeuova, Amantay Iskanderov, Saltanat Zhokizhanova

**Affiliations:** 1Department of Inorganic and Technical Chemistry, Karaganda Buketov University, Karaganda 100028, Kazakhstan; dr.amantay@ya.ru; 2Graduate School of Science, Astana International University, Astana 020000, Kazakhstan; abeuova.salta@gmail.com; 3Department of Physics and Chemistry, Saken Seifullin Kazakh Agro Technical Research University, Astana 010000, Kazakhstan; saltanat_zh75@mail.ru

**Keywords:** azulene, π-conjugated azulenes, aniline azulenes, cross-coupling, absorption spectra, fluorescence spectra

## Abstract

In this work, through the Suzuki–Miyaura cross-coupling reaction with high yields, new π-conjugated azulene compounds containing diphenylaniline groups at positions 2 and 6 of azulene were synthesized. The obtained diphenylaniline–azulenes have intensely visible-light absorbing and emitting (in the wavelength range from 400 to 600 nm) properties. It has been shown that such unique optical properties, in particular fluorescent emission in the region of blue and green photoluminescence (λem at 495 and 525 nm), which were absent in the original azulene, are the result of the electron donor effect of diphenylaniline groups, which significantly changes the electronic structure of azulene and leads to the allowed HOMO → LUMO electron transition.

## 1. Introduction

The growing interest in aromatic compounds with an extended π-electron conjugation system is due to their importance as functional materials for organic optoelectronics.

Currently, much attention is paid to arylated, as well as aromatic and heteroaromatic, compounds substituted with electron acceptor and/or electron donor groups.

For example, diarylamine-substituted naphthalene and anthracenes, which have excellent electron donor properties, have attracted great interest due to their potential in organic semiconductor materials, redox materials, metal–organic framework (MOF) structures, and organic light-emitting diodes (OLEDs) [1,2,3,4,5,6,7,8,9,10].

One advantage of such aromatic systems is that they can fine-tune the electronic structure of materials in order to optimize performance and morphology.

In this regard, the structural isomer of naphthalene azulene [11,12,13,14,15,16,17,18,19,20] is of considerable interest. The non-alternating aromatic hydrocarbon azulene **1** has unique electronic and spectral properties, including a polarized structure with a dipole moment of the order of 1.08 D [21] and anomalous anti-Kasha fluorescence S_2_ → S_0_ [22,23,24,25,26].

The structure of azulene can be considered a tropylium cation condensed with a cyclopentadienyl anion (Figure 1a). This non-alternating and polarized azulene structure results in high energy levels of HOMO frontal orbitals and low energy levels of LUMO orbitals compared to other conventional aromatic hydrocarbons [11,12,13,14,15]. In azulene, positions 1 and 3 have large HOMO coefficients, and positions 2 and 6 have large HOMO-1 and LUMO coefficients (Figure 1b) [11,12,13,14,15]. Besides, azulene, unlike its colorless isomer naphthalene, has a blue color and shows the absorption caused by the transition of S_0_–S_1_ with an λmax of about 580 nm [27]. However, for the allowed π-π*-transition, the molar coefficient of this absorption is very small and is only 350 M^−1^ cm^−1^ [27].

Thus, the introduction of different functional groups into the azulene structure, in particular electron donor groups at positions 2 and 6, can lead to significant changes in its electronic structure and give new functional materials with unique photophysical properties.

Here, we report the synthesis of novel π-conjugated azulene compounds **4** and **6** containing electron donor diphenylaniline groups at positions 2 and 6 via Suzuki–Miyaura cross-coupling. Also, in the study of their optical properties and redox behavior, we show that similar molecular constructs of azulene allow the electronic transition of HOMO → LUMO and lead to the appearance of new intense absorption and emission bands in the visible region of the spectrum.

## 2. Results and Discussion

Synthetic routes leading to azulene π-conjugate compounds: 2-(N,N-diphenylaniline)-azulene **4** and 2,6-bis(N,N-diphenylaniline)-azulene **6** are presented in Figure 1 and Figure 2.

As can be seen from the first scheme, the key molecule 2-(4,4,5,5-tetramethyl-1,3,2-dioxaborolanyl)-azulene **2** was obtained by the direct C_2_-H borylation of azulene **1** with Bis (pinacolato) diboron using an iridium catalyst according to the procedure [28]. Further, the Suzuki–Miyaura coupling between 4-bromotriphenylamine **3** and borylazulene **2** in THF/water mixture (4:1) in the presence of the Pd(PPh_3_)_2_CI_2_ catalyst gives the final product **4** in a high 86% yield.

Another key molecule, 2,6-bis(4,4,5,5-tetramethyl-1,3,2-dioxaborolanyl)-azulene **5,** was synthesized by the borylation reaction of the 6 position of 2-borylazulene **2,** also in the presence of an iridium catalyst according to procedure [29] (Figure 2). Similarly, the Suzuki–Miyaura coupling reaction between amine **3** and 2,6-diborylazulene **5** results in the final diphenylaniline–azulene **6** in a high yield of 83%.

The obtained diphenylaniline–azulenes **4** and **6** are dark green and red–brown substances (unlike the blue color of the original azulene), which are completely soluble at room temperature in organic solvents such as dichloromethane, chloroform, and chlorobenzene.

The chemical structure and purity of the synthesized compounds 2-(N,N-diphenylaniline)-azulene **4** and 2,6-bis(N,N-diphenylaniline)-azulene **6** were proven by NMR (^1^H and ^13^C), IR, and mass spectrometry as well as elemental analysis (see Appendix A).

The electron spectra of diphenylaniline–azulene **4** and **6** in the UV-Vis range in dichloromethane (DCM) at room temperature are presented in Figure 2, and the corresponding data are summarized in Table 1.

To assess the electron donor effect of the diphenylaniline groups and the efficiency of π-conjugation, the absorption spectra of compounds **4** and **6** were compared with the spectrum of the original azulene **1** (Figure 2). As shown in Figure 2, the monosubstituted diphenylaniline azulene **4** at position 2 shows a new broad absorption band in the visible spectrum with the maximum at 436 nm and molar absorption coefficient of 24,637 M^−1^ cm^−1^ (Table 1).

Diphenylaniline–azulene **6,** disubstituted at positions 2 and 6, also demonstrates a new strong absorption band in the visible range, with a maximum at 468 nm and molar coefficient of 86,888 M^−1^ cm^−1^ (Table 1). As shown in Figure 2, the visible absorption maximum **6** is shifted to the red region at 32 nm and has an intensity several times higher than the absorption maximum of the monosubstituted compound **4**.

Such changes can occur as a result of a twofold expansion of the π-conjugation and an increase in electron delocalization (Figure 3), leading to a decrease in the energy gap of HOMO-LUMO molecule **6**.

In addition, the electron absorptions in the visible spectrum of compounds **4** (ε 24,637 M^−1^ cm^−1^) and **6** (ε 86,888 M^−1^ cm^−1^) are much stronger than those of the original azulene **1** (ε 350 M^−1^ cm^1^) (Table 1) [27].

It should be noted that the visible absorption bands **4** and **6** are at the same wavelength (400–600 nm) as a number of commercially available functional materials used in organic solar cells (e.g., oligothiophenes, indophenines) [30,31,32,33,34] and organic dyes [35,36,37].

The fluorescence spectra of diphenylaniline–azulenes **4** and **6** in DCM at room temperature are presented in Figure 4, and the corresponding data are summarized in Table 1.

As can be seen from Figure 4, the monosubstituted compound **4** shows a new broad emission band in the visible region, with the maximum at 495 nm (with excitation at the wavelength of 425 nm) (Table 1).

Disubstituted molecule **6** also shows a new intense emission band in the visible spectrum, with the maximum at 523 nm (with excitation at the wavelength of 425 nm) (Table 1). From Figure 4, it can be seen that the maximum of the visible emission of **6** is shifted to the red region at 30 nm and has a higher intensity than the fluorescence emission of compound **4** (Table 1). The red shift of the fluorescent emission band with increasing intensity can also occur as the result of expansion of π-conjugation (Figure 3), leading to a decrease in the energy gap between the HOMO and LUMO of compound **6.**

The ability of π-conjugated compounds **4** and **6** to intensely emit visible light in the region of blue and green photoluminescence is unique, since the original azulene **1** does not possess this property (Table 1).

Thus, it has been shown that by introducing electron donor diphenylaniline groups into the azulene structure, namely at positions 2 and 6, unique strong electron absorbances (ε 24,637 M^−1^ cm^−1^ and ε 86,888 M^−1^ cm^−1^) and fluorescent emission (257 and 264 a.u) are induced in the visible region of the spectrum.

To understand the electronic structure of the obtained diphenylaniline–azulenes **4** and **6**, and the relationship between structure and optical property, calculations were performed using density functional theory (DFT) with the B3LYP/6- 31G * functional (Figure 5, see Appendix A).

As can be seen from Figure 5, the frontal orbitals of HOMO π-conjugated compounds **4** and **6** are allocated both along the azulene skeleton and along the diphenylaniline group. This distribution of orbitals may result from the loosening interaction between the HOMO-1 of azulene and the HOMO of N,N-diphenylaniline [38] because the carbon atoms 2 and 6 are in the nodal plane in the HOMO of azulene, while in HOMO-1 they have large coefficients (Figure 1b).

In addition, it is shown that the HOMO of diphenylaniline–azulenes **4** (−4.85 eV) and **6** (−4.74 eV) are located higher in level than the HOMO of the original azulene **1** (−5.19 eV) and have a reduced HOMO-LUMO energy gap by 0.42 and 0.58 eV (Figure 5), apparently due to the inversion of the order of energy levels of molecular orbitals between the original compound **1** and diphenylaniline–azulenes **4** and **6** [38].

As a result of such changes in the electronic structure of azulene, the previously forbidden electronic transition HOMO → LUMO becomes allowed [38] and, as a consequence, leads to unique intense absorption and emission in the visible region, which we actually observe from the absorption and fluorescence spectra of compounds **4** and **6** (Figure 2 and Figure 4, Table 1).

To determine the electrochemical properties, the redox behaviors of diphenylaniline–azulenes **4** and **6** were investigated by cyclic voltammetry (CV) (Figure 6). Measurements were made with a standard three-electrode configuration (see Appendix A).

As shown in Figure 6, the monosubstituted compound **4** exhibits reversible double oxidation peaks at 0.46 V and 0.75 V, respectively. In addition, molecule **4** shows an irreversible reduction peak at −1.86 V. Disubstituted molecule **6** shows an irreversible single oxidation peak at 0.65 V, as well as an irreversible reduction peak at −1.85 V.

According to the onset of the oxidation peak (0.32 V for Eox^onset^) and the onset of the reduction peak (−1.65 V for Ered^onset^), we can conclude that the corresponding HOMO and LUMO energy levels of molecule **4** are −4.88 eV and −2.75 eV, respectively. Also, with the onset of oxidation peak (0.38 V for Eox^onset^) and the onset of reduction peak (−1.68 V for Ered^onset^), the HOMO and LUMO energy levels of compound **6** are −4.82 eV and −2.72 eV, respectively. Energy levels were calculated using the following formula: E_HOMO_ = −4.80 eV − [(Eox^onset^) − E_1/2_(ferrocene)] and E_LUMO_ = −4.80 eV − [(Ered^onset^) − E_1/2_(ferrocene)] [39].

It should be noted that the energy levels of HOMO diphenylaniline–azulenes **4** and **6** obtained as a result of electrochemical studies are close to those calculated on the basis of theoretical modeling (DFT calculations, Figure 5).

The theoretical values of LUMO energy levels of compounds **4** and **6** are slightly different from the experimental ones, resulting in an approximate difference of 0.85 V and 0.76 V, respectively, in the band gap obtained using these two methods [39].

## 3. Materials and Methods

NMR spectra of ^1^H and ^13^C were recorded on a JNM-ECA 500 spectrometer in CDCl_3_ at room temperature using tetramethylsilane (TMS) as an internal standard. The NMR spectrometer operating frequencies were 500 MHz (^1^H) and 126 MHz (^13^C). IR spectra were recorded on an Avatar-360 Fourier spectrometer in KBr tablets. Mass spectra (EI) were determined with an Agilent 6530 Accurate-Mass Q-TOF LC/MS system. Elemental analysis was performed on a CHNS-O UNICUBE-elemental analyzer. The melting points were determined on a Melting Point M-560 apparatus.

Absorption spectra were recorded on a Shimadzu UV-1800 spectrophotometer. Fluorescence spectra were recorded on an Agilent Cary Eclipse fluorescence spectrophotometer. Cyclic voltammetry (CV) measurements were performed on a PalmSens electrochemical analyzer.

Commercially available reagents and solvents azulene, 4-bromotriphenylamine, Bis(pinacolato)diboron, [IrCl(cod)]_2_, Pd(PPh_3_)_2_CI_2_, 2,2′-Bipyridine, cyclohexane, THF, and others were used as received. 2-(4,4,5,5-tetramethyl-1,3,2-dioxaborolanyl)-azulene **2** and 2,6-Bis(4,4,5,5-tetramethyl-1,3,2-dioxaborolanyl)-azulene **5** were prepared according to the procedure [28,29].

2-(N,N-diphenylaniline)-azulene (**4**). Pd(PPh_3_)_2_CI_2_ (16 mg, 0.02 mmol) and K_2_CO_3_ (118 mg, 0.86 mmol) were added in argon to the mixture of 4-bromotriphenylamine **3** (140 mg, 0.43 mmol) and 2-(4,4,5,5-tetramethyl-1,3,2-dioxaborolanyl)-azulene 2 (220 mg, 0.86 mmol) in 10 mL degassed THF/H_2_0 (4:1). The mixture was stirred for 18 h at 75 °C, and then cooled to room temperature and extracted with CH_2_Cl_2_ (3 × 18 mL). The combined extracts were dried over MgSO_4_ and evaporated in vacuo. The product was purified by column chromatography (on silica gel) with the mixture of C_6_H_14_/CH_2_Cl_2_ (9:1) and recrystallization from C_6_H_14_/CH_2_Cl_2_ to give a dark brown solid (274 mg, 86% yield). M.p. 168–170 °C. IR (ν, cm^−1^): 2924, 2853, 1586,1486,1405,1327, 1225, 1175, 1073, 696. ^1^H NMR: δ 8.24 (d, J = 9.4 Hz, 1H), 7.84 (d, J = 8.3 Hz, 1H), 7.62 (s, 1H), 7.47 (s, 1H), 7.29–6.94 (m, 17H). ^13^C NMR: δ 147.94, 147.53, 147.54, 141.54, 135.78, 135.27, 132.24, 129.48, 129.31, 128.51, 125.23, 124.88, 124.50, 124.26, 123.86, 123.82, 123.44, 123.36, 123.32, 122.71, 113.95. MS (EI), *m*/*z*: 371.1 [M]^+^. C_28_H_21_N: calcd. C 90.53, H 5.70, N 3.77; found C 90.26, H 5.47, N 3.84.

2,6-Bis(N,N-diphenylaniline)-azulene (**6**). Pd(PPh_3_)_2_CI_2_ (16 mg, 0.02 mmol) and K_2_CO_3_ (60 mg, 0.43 mmol) were added in argon to the mixture of 4-bromotriphenylamine **3** (560 mg, 1.72 mmol) and 2,6-bis(4,4,5,5-tetramethyl-1,3,2-dioxaborolanyl)-azulene **5** (163 mg, 0.43 mmol) in 10 mL degassed THF/H_2_0 (4:1). The mixture was stirred for 18 h at 75 °C, and then cooled to room temperature and extracted with CH_2_Cl_2_ (3 × 18 mL). The combined extracts were dried over MgSO_4_ and evaporated in vacuo. The product was purified by column chromatography (on silica gel) with the mixture of C_6_H_14_/CH_2_Cl_2_ (9:1) and recrystallization from CH_2_Cl_2_ to give a red–brown solid (225 mg, 83% yield). M.p. 213–215 °C. IR (ν, cm^−1^): 2924, 2856, 1588, 1486, 1411, 1322, 1277, 1176, 1075, 695. H NMR: δ 8.24 (d, *J* = 10.3 Hz, 2H), 7.84 (d, *J* = 8.6 Hz, 2H), 7.58 (s, 1H), 7.51 (s, 1H), 7.38 (dd, *J* = 9.9, 3.5 Hz, 4H), 7.38–7.36 (m, 4H), 7.31–7.26 (m, 8H), 7.17–7.14 (m, 16H). ^13^C NMR: δ 149.67, 148.13, 147.94, 147.53, 147.46, 147.11, 141.54, 135.78, 137.27, 132.24, 130.26, 129.48, 129.45, 129.31, 128.51, 125.85, 125.50, 125.23, 124.88, 124.26, 123.86, 13.82, 123.44, 123.36, 123.32, 122.76, 114.85, 113.95. MS (EI), *m*/*z*: 614.2 [M]^+^. C_46_H_34_N_2_: calcd. C 89.86, H 5.56, N 4.57; found C 89.56, H 5.42, N 4.64.

## 4. Conclusions

Thus, for the first time, we have successfully synthesized intensely visible-light absorbing and emitting π-conjugated azulene compounds: 2-(N,N-diphenylaniline)-azulene **4** and 2,6-bis(N,N-diphenylaniline)-azulene **6**. Such unique optical properties, in particular fluorescent emission of visible light in the region of blue and green photoluminescence, which were absent in the original azulene, were achieved as the result of the introduction of electron donor diphenylaniline groups in the 2 and 6 positions of azulene, leading to significant changes in its electronic structure and the appearance of an allowed HOMO → LUMO electronic transition.

The synthesized azulene compounds have higher HOMO orbitals compared to the oligothiophenes used in electronic devices and exhibit strong absorption properties over the wide wavelength range.

The results obtained provide a rational approach to the design of a representative series of new π-conjugated azulene-based compounds for optoelectronic and photonic applications.

## Data Availability

The original contributions presented in the study are included in the article, further inquiries can be directed to the corresponding authors.

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
