# Peer review of "Introduction of Electron Donor Groups into the Azulene Structure: The Appearance of Intense Absorption and Emission in the Visible Region"

_molecules, 2024, doi:10.3390/molecules29143354_

Round 1

Reviewer 1 Report

Comments and Suggestions for Authors

The reviewed paper presents the synthesis and the selected properties of two compounds prepared based on azulene. This manuscript shows interesting properties of these compounds such as fluorescent emission. However, the range of compounds tested is very limited. The authors describe only two compounds. . Have substituted derivatives of such compounds been tested? If so, what properties do they have? Moreover, I have a few questions before possible publication of this paper.

1. I understand that these compounds are crystalline solids. Are these compounds stable at the melting point or rather they decompose at the melting point?

2. What about the chemical resistance of these compounds? Are they resistant to moisture?

3. Are these compounds thermally resistant? 

Author Response

Dear Reviewer,

Thank you for your review. Your recommendations and comments helped significantly improve the article. We haverevised the article in accordance with your recommendations:

Comment 1: The reviewed paper presents the synthesis and the selected properties of two compounds prepared based on azulene. This manuscript shows interesting properties of these compounds such as fluorescent emission. However, the range of compounds tested is very limited. The authors describe only two compounds. Have substituted derivatives of such compounds been tested? If so, what properties do they have? Moreover, I have a few questions before possible publication of this paper.

Response 1: Thank you for point this out.  Considering that the task was set in the work to obtain exactly such a molecular structure where the azulene ring is conjugated with a p-phenylene group, which gives intense absorption and emission in the visible region (due to inversion of HOMO levels and obtaining a permitted π, π transition), the synthesis of other derivatives was not considered.

Comment 2:  I understand that these compounds are crystalline solids. Are these compounds stable at the melting point or rather they decompose at the melting point?

Response 2: Thank you for point this out.  In determining the melting point of compounds 4 and 6 (Melting Point M-560), the beginning of melting (i.e. softening of the substance and melting of the first crystals) and the end of melting (i.e. melting of the last crystals of the substance and complete transition to a transparent liquid) were clearly observed. The determination was carried out with three times the repeatability. No degradation was observed, and the resulting compounds were stable at the melting point. 

Comment 3:  What about the chemical resistance of these compounds? Are they resistant to moisture?

Response 3: Thank you for point this out.  No changes (i.e., color, solubility, spectroscopic data, etc.) occurred in the air during operation with compounds 4 and 6 for the entire extended period of time, and the compounds were stable

 Comment 4: Are these compounds thermally resistant? 

Response 4: Thank you for point this out.  Since the obtained compounds 4 and 6 are not polymers, the determination of their heat resistance by methods such as TGA was not carried out.

Reviewer 2 Report

Comments and Suggestions for Authors

The article titled “Introduction of electron-donor groups into the azulene structure: the appearance of intense absorption and emission in the visible region” discusses the topic of azulene derivatives. The authors do not clearly present the thesis of the research undertaken. Which is a definite downside. Below are my comments:

1. The nomenclature should be corrected and unified.

2. The measurement conditions for electrochemical tests are not listed anywhere. This should be added.

3. Are you sure that the results obtained from CV measurements are reversible?

4. Please correct the reduction results in the table. They are misinterpreted.

5. "Comparison of the first oxidation potentials 4 (0.46 V) and 6 (0.65 V) (Table 2) shows that the 2,6-disubstituted compound 6 possesses stronger electron-donor properties than the monosubstituted molecule 4." This sentence is not well written. As we know, molecule 4 with a potential of 0.46 V is more easily oxidized. Please correct.

Author Response

Dear Reviewer,

Thank you for your review. Your recommendations and comments helped significantly improve the article. We haverevised the article in accordance with your recommendations:

Comment 1: The authors do not clearly present the thesis of the research undertaken. Which is a definite downside.

Response 1: Thank you for point this out.  We agree with this comment. The article abstract is supplemented with values ​of absorption and fluorescence wavelengths. (lines 15,17)

Comment 2: The nomenclature should be corrected and unified.

Response 2: Thank you for point this out.  We agree with this comment. The nomenclature of the obtained diphenylaniline-azulenes 4 and 6 is unified and corrected. (lines 69,70,72,78,87, 88, 249, 250, 252, 253, 265, 282,2 83)

Comment 3: The measurement conditions for electrochemical tests are not listed anywhere. This should be added.

Response 3: Thank you for point this out.  We agree with this comment. The conditions for measuring the electrochemical properties of diphenylaniline-azulenes 4 and 6 have been added to the supplementary materials of the article (Supplementary Materials , section 3).

Comment 4:  Are you sure that the results obtained from CV measurements are reversible?

Response 4: Thank you for point this out.  We agree with this comment. After careful analysis of the results of CV measurements, it was found that molecule 4 undergoes reversible oxidation and irreversible reduction. Compound 6 - irreversible oxidation and irreversible reduction. Changes made. (lines 210-227).

Comment 5:  Please correct the reduction results in the table. They are misinterpreted.

Response 5: Thank you for point this out.  We agree with this comment. The results of cyclic voltammetry (CV) were carefully analyzed and revised. Changes made. (lines 210-227). 

Comment 6:  "Comparison of the first oxidation potentials 4 (0.46 V) and 6 (0.65 V) (Table 2) shows that the 2,6-disubstituted compound 6 possesses stronger electron-donor properties than the monosubstituted molecule 4." This sentence is not well written. As we know, molecule 4 with a potential of 0.46 V is more easily oxidized. Please correct.

Response 6: Thank you for point this out.  We agree with this comment. The results of cyclic voltammetry (CV) were carefully analyzed and revised. Changes made. (lines 210-227). 

Reviewer 3 Report

Comments and Suggestions for Authors

Details of the DFT for calculating the HOMO and LUMO energy levels and molecular orbital diagrams need to be added, e.g. what units were used and whether there was London dispersion.

Regarding planarity or torsion angles I am interested, is there a crystal structure? If so please add and resolve. If not please calculate the dihedral angles between aromatic rings using DFT simulations.

Based on the oxidation and reduction peaks obtained from CV electrochemical tests the HOMO and LUMO energy levels can be calculated, please add these results. Also compare with the energy levels calculated from theoretical simulations to draw new conclusions.

The following article on donor-acceptor I suggest to refer to further improve the readability of the article:1.Synthesis and Characterization of Solution-Processed Indophenine Derivatives for Function as a Hole Transport Layer for Perovskite Solar Cells. 2. Donor-Acceptor-Based Organic Polymer Semiconductor Materials to Achieve High Hole Mobility in Organic Field-Effect Transistors

Author Response

Dear Reviewer,

Thank you for your review. Your recommendations and comments helped significantly improve the article. We haverevised the article in accordance with your recommendations:

Comment 1: Details of the DFT for calculating the HOMO and LUMO energy levels and molecular orbital diagrams need to be added, e.g. what units were used and whether there was London dispersion.

Response 1: Thank you for point this out.  We agree with this comment. DFT details for HOMO and LUMO energy levels and molecular orbital plots, including units used, are added to the article (Supplementary Materials, Section 2).

Since single molecules 4 and 6 were studied, the London dispersion was not used in the calculations.

Comment 2: Regarding planarity or torsion angles I am interested, is there a crystal structure? If so please add and resolve. If not please calculate the dihedral angles between aromatic rings using DFT simulations.

Response 2: Thank you for point this out.  We agree with this comment. The dihedral angles of compounds 4 and 6 are calculated using DFT simulation. As can be seen from Figures 1 and 2 (please see the attached file), the nitrogen atom of each diphenylaniline group of molecules 4 and 6 has a plane structure, and the dihedral angles between the sides of the 5-membered and 7-membered azulene rings differ significantly, at 30 and 30°, respectively.

Comment 3: Based on the oxidation and reduction peaks obtained from CV electrochemical tests the HOMO and LUMO energy levels can be calculated, please add these results. Also compare with the energy levels calculated from theoretical simulations to draw new conclusions.

Response 3: Thank you for point this out.  We agree with this comment. The article in the CV part of the studies was finalized according to the comments and all results with new conclusions were added to the article (lines 210-227)   

Comment 4: The following article on donor-acceptor I suggest to refer to further improve the readability of the article: 1.Synthesis and Characterization of Solution-Processed Indophenine Derivatives for Function as a Hole Transport Layer for Perovskite Solar Cells. 2. Donor-Acceptor-Based Organic Polymer Semiconductor Materials to Achieve High Hole Mobility in Organic Field-Effect Transistors.

Response 4: Thank you for point this out.  We agree with this comment. To improve the readability of the paper, we have referenced the following relevant works: 1. Synthesis and Characterization of Solution-Processed Indophenine Derivatives for Function as a Hole Transport Layer for Perovskite Solar Cells 2. Donor-Acceptor-Based Organic Polymer Semiconductor Materials to Achieve High Hole Mobility in Organic Field-Effect Transistors (line 125, positions [33] and  [34] in the Reference list)

Round 2

Reviewer 2 Report

Comments and Suggestions for Authors

All comments have been corrected.